# Antimicrobial Treatment of *Staphylococcus aureus* Biofilms

**DOI:** 10.3390/antibiotics12010087

**Published:** 2023-01-04

**Authors:** Felipe Francisco Tuon, Paula Hansen Suss, Joao Paulo Telles, Leticia Ramos Dantas, Nícolas Henrique Borges, Victoria Stadler Tasca Ribeiro

**Affiliations:** 1Laboratory of Emerging Infectious Diseases, School of Medicine, Pontifícia Universidade Católica do Paraná, Curitiba 80215-901, Paraná, Brazil; 2AC Camargo Cancer Center, Infectious Diseases Department, São Paulo 01525-001, São Paulo, Brazil

**Keywords:** antibiotic, biofilm, infections, *Staphylococcus aureus*, therapeutic antibiofilm

## Abstract

*Staphylococcus aureus* is a microorganism frequently associated with implant-related infections, owing to its ability to produce biofilms. These infections are difficult to treat because antimicrobials must cross the biofilm to effectively inhibit bacterial growth. Although some antibiotics can penetrate the biofilm and reduce the bacterial load, it is important to understand that the results of routine sensitivity tests are not always valid for interpreting the activity of different drugs. In this review, a broad discussion on the genes involved in biofilm formation, quorum sensing, and antimicrobial activity in monotherapy and combination therapy is presented that should benefit researchers engaged in optimizing the treatment of infections associated with *S. aureus* biofilms.

## 1. Introduction

*Staphylococcus aureus* is a Gram-positive bacterium associated with several diseases in community- and hospital-acquired settings. It is among the most common opportunistic human pathogens and is associated with a high rate of morbidity and mortality [1]. The colonization rate of the bacterial population is approximately 30%, in general, in the skin and nasal mucosa [2]. *S. aureus* infects almost all tissues of the body; the infections include bacteremia, infective endocarditis, skin and soft tissue infections, osteomyelitis, arthritis, pneumonia, empyema, meningitis, and urinary tract infections. *S. aureus* can also cause toxin-mediated clinical syndrome, which includes gastroenteritis and toxic shock syndrome (Figure 1) [3,4]. Hospitalizations related to staphylococcal infections are frequent and commonly associated with high mortality rates and increasing health costs [5,6]. In this setting, surgical site infections and biofilm-associated infections (those associated with implantation of medical devices) are common.

*S. aureus* is associated with several virulence factors, including surface proteins involved in bacterial adherence, extracellular enzymes and toxins that promote tissue necrosis, and factors that interfere with the immune system [7]. *S. aureus* can synthesize some enzymes to enhance its pathogenicity and dissemination within the host; these include coagulase, hyaluronidase, deoxyribonuclease, and lipase [8]. Extracellular protein toxins, including enterotoxins, toxic shock syndrome toxin 1 (TSST-1), exfoliative toxins (ETs), hemolysins, epidermal cell differentiation inhibitors (EDINs), and Panton–Valentine leukocidin (PVL), can also enhance the pathogenicity [9]. The process of *S. aureus* infections can involve the following progressive stages: (1) colonization, (2) local infection, (3) systemic dissemination and/or sepsis, (4) metastatic infections, and (5) toxinosis. *S. aureus* biofilm formation, toxin production, and immune evasion strategies limit antibacterial immune responses of the host [10].

*S. aureus* is among the most difficult-to-treat bacteria, which include *Enterococcus faecium*, *Klebsiella pneumoniae*, *Acinetobacter baumannii*, *Pseudomonas aeruginosa*, and *Enterobacter* species. These bacteria belong to the ESKAPE pathogen group that warrants the urgent development of new therapies in view of multidrug resistance [11]. Overall, staphylococci are the leading causes of device-related infections [12]. Among staphylococci, *S. aureus* is of most clinical concern because it is commonly associated with more severe and aggressive infections. The higher severity of *S. aureus* infections is due to its biofilm-forming ability and production of several toxins [13,14].

Medical devices particularly prone to infection include intravascular (central venous catheters, mechanical heart valves, and pacemakers) and extravascular (endotracheal tubes, intrauterine devices, peritoneal dialysis catheters, prosthetic joints, and others) implants [15]. After the insertion of a device, *S. aureus* can colonize it during implantation or during an asymptomatic or symptomatic bacteremia. With colonization, there is a biofilm formation through attachment of bacterial cells to human matrix proteins, including fibronectin and fibrinogen. These human proteins commonly cover the devices soon after insertion [16]. There are several reviews on staphylococcal biofilm [17,18,19], but the approach to antimicrobial therapy is not in-depth, whether reviewing monotherapy and combination therapy applied to real life settings, bringing translational medicine from in vitro studies to clinical application. The biofilm enables persistence of the microorganism and renders the bacteria with increased antibiotic tolerance and immune evasion properties. A variety of factors contribute to this critical process and are discussed in the following topics, focused on antimicrobial therapy as monotherapy or combination therapy.

## 2. Virulence Factors

Several virulence factors that help in colonization, dissemination, and immune evasion are involved in *S. aureus* infection. These factors are used for attachment of bacteria to host cells, breakdown of the host immunity, and tissue invasion, which may lead to sepsis, as well as for eliciting toxin-mediated syndromes. This is the basis for persistent staphylococcal infections without a strong host immune response [20,21]. Virulence factors of *S. aureus* include those that aid attachment of bacterial cells to host tissues [22], breaking/evading the host immunity [23], tissue invasion [24], and induction of toxicosis [25,26].

## 3. Antibiotic Resistance

According to the 2022 World Health Organization (WHO) report, antimicrobial resistance (AMR) is prevalent and can impact individuals of any age, in any country of the world. The consequences of unchecked AMR are wide-ranging and extremely costly, not only financially, but also in terms of global health, food security, environmental well-being, and socioeconomic development. Enzymatic hydrolysis, enzymatic modification of antibiotics by group transfer and redox process, modification of antibiotic targets, reduced permeability to antibiotics by modification of porins, and active extrusion of antibiotics by membrane efflux pumps are the most common cellular mechanisms underlying antibiotic resistance [27,28].

*S. aureus* has acquired resistance to several antibiotics rapidly. This resistance poses a significant problem in the treatment of *S. aureus* infections in humans. Moreover, *S. aureus* produces antibiotic-neutralizing enzymes, resulting in numerous mechanisms of resistance to therapeutic drugs [29]. Enzymes involved in antibiotic resistance play a significant role in bacterial resistance to antibiotic pressure regarding diversity, evolution, and spread. Antibiotic-producing bacteria need strategies to counteract the deadly effects of chemicals by producing degradative enzymes. However, the selection pressure caused by the widespread use of antibiotics in humans, animals, and the environment has resulted in the propagation of resistant bacterial clones [30].

The ability to acquire resistance to multiple antibiotics makes *S. aureus* a challenging pathogen to treat. Methicillin-resistant *S. aureus* (MRSA) and vancomycin-resistant *S. aureus* (VRSA) are the most common types of antibiotic-resistant *S. aureus* strains. Currently, only MRSA is categorized as an agent of high significance with the potential to cause considerable worldwide mortality in the absence of effective containment and treatment options [31,32]. The increased prevalence of multidrug-resistant forms, which include VRSA, is a major problem. The annual mortality from diseases caused by antibiotic-resistant bacteria has surpassed 10 million and is expected to outnumber cancer deaths by 2050 [33].

Mobile genetic elements (MGEs) play an integral part in the adaptation of *S. aureus* to environmental stresses, which include antibiotic exposure. MGEs are the primary means by which genetic information is exchanged between bacteria via horizontal gene transfer. *S. aureus* strains, in general, contain a relatively large variety of MGEs, including plasmids, transposons, bacteriophages, pathogenicity islands, and staphylococcal cassette chromosomes. Staphylococcal cassette chromosomes have played a central role in conferring resistance to β-lactam antibiotics and vancomycin [34].

## 4. Toxins

*S. aureus* produces several secreted and surface-bound proteins that enable it to attach to host tissues as well as toxins. The pathogenicity of *S. aureus* is attributed to its virulence factors, and toxins play an important role in the process [14]. These toxins are categorized into the following three major groups: (1) the pore-forming toxins (PFTs), (2) exfoliative toxins (ETs), and (3) superantigens (Sags). PFTs are divided into the following four types: (1) hemolysin-α, (2) hemolysin-β, (3) leukotoxins, and (4) phenol-soluble modulins (PSMs) [35]. The Agr quorum sensing system regulates the *S. aureus* toxins through a variety of metabolic adaptations [36]. PSMs, which are the most-produced PFTs, are exceptionally regulated by direct binding of the AgrA response regulator to psm operon promoters [37]. PSMs are completely absent in mutants with a dysfunctional Agr system [38].

*S. aureus* toxins are related to the pathogenesis of some diseases, such as toxic shock syndrome (TSS), staphylococcal scalded skin syndrome, necrotizing pneumonia, and deep-seated skin infections [39]. The toxins can damage the cell membranes of the host, either by degrading intercellular connections or by modulating immune responses [35]. Toxins are attractive key targets for innovative therapeutics. Among the possible therapeutic strategies, the use of neutralizing antibodies and vaccines are the most promising ones [40].

## 5. Biofilms

The presence of a suitable substrate is the most crucial prerequisite for biofilm formation. The condition and type of surface are the key determinants. Bacteria frequently colonize rough surfaces. For this reason, the potential for biofilm development is different for distinct materials, such as glass, metals, and plastic polymers. The rate and extent of adherence of bacteria to the surface vary depending upon the chemical composition that coat the biofilm [41].

The notorious ability of *S. aureus* to form biofilm needs to be highlighted [42]. This structure, comprised of an extracellular polymeric substance (EPS), not only helps bacteria resist the environment, but is also important from the perspective of the action of antimicrobials because it hinders the penetration of drugs and allows for host immune evasion [43], contributing to bacterial virulence [44].

The biofilm development is a complex process, which can be didactically divided into the following steps: (i) attachment, (ii) extracellular matrix synthesis with bacterial proliferation, (iii) biofilm structuring, and (iv) cell detachment (Figure 2). *S. aureus* attaches to a surface (biotic or abiotic) using different mechanisms, involving adhesins [45,46], teichoic acids [47,48], changes in hydrophobicity of bacterial cell surface [49], and extracellular DNA production [50]. After the attachment, the bacterial cells proliferate on the surface, forming microcolonies. In the next stage, the biofilm is established with the development of microcolonies. During proliferation and biofilm maturation, the bacterial cells are held together by adhesive factors in the new matrix. Remodeling of the biofilm occurs through disruptive factors, which include surfactants and nucleases, which are essential for the three-dimensional structure of the mature biofilm with its distinctive towers and channels. During biofilm disassembly, cell detachment is mainly caused by a protease-driven degradation of the biofilm matrix and disruption of the biofilm by PSMs [51,52].

The EPS matrix that remains after the death of bacteria caused by an antibiotic treatment or the immune system can promote recolonization of the surface, either by the same or other bacterial species, causing recurrence of infection and further complications [53]. Consequently, for effective management of biofilm infections, removal of the residual EPS matrix can be as crucial as killing the bacteria. Additionally, considering the variability in the composition of *S. aureus* EPS matrix and interaction between its multiple components, strategies to disrupt the matrix should ideally target several constituents of the matrix simultaneously [54]. In *S. aureus* biofilms, poly-*N*-acetyl-β-(1–6)-glucosamine (PNAG), proteins and extracellular DNA are the main components of the EPS matrix. PNAG helps in the formation of biofilms and protects bacteria from the host immune system [55]. Adhesion to the surface and initiation of biofilm formation are also related to the expression of numerous proteins, such as cell wall-anchored proteins, phenol soluble modulins, and recycled cytoplasmic proteins found in both MSSA and MRSA [45].

## 6. Biofilm Formation on Medical Devices

The structural complexity of biofilm enables bacterial growth on numerous surfaces. Artificial devices provide a fertile ground for the establishment of bacteria. The biofilms formed by *S. aureus* on medical devices, such as prosthetics, contact lenses, urinary and central venous catheters, endotracheal tubes, and artificial heart valves, pose a great challenge as they are associated with chronic infections [31]. Usually, an indwelling device or implant readily becomes coated with the host matrix, which eventually allows for the initial attachment of bacteria and subsequent formation of a biofilm [56]. Cross-contamination during surgery is a major source of device-related infections [57]. Not surprisingly, immunocompromised patients, such as AIDS patients, patients receiving immunosuppressive therapy, or premature newborns, are at higher risk of developing biofilm-associated infections on indwelling devices [58]. These populations are also at increased risk of serious complications arising from infected devices. Notably, this structure is associated with increasing rates of antimicrobial drug resistance and serious clinical implications [59], warranting novel therapeutic strategies. The adhesion of *S. aureus* biofilm on a medical device, as visualized using scanning electronic microscopy (SEM), is shown in Figure 3. Implant-associated staphylococcal biofilm infections normally require surgical debridement and prolonged antimicrobial therapy [60].

## 7. Quorum Sensing

Quorum sensing (QS) bacteria produce signaling molecules called autoinducers, which at stimulatory concentrations play a role in gene alteration [61]. Biofilm formation is a behavior of the social group and is segregated into different processes of composition—each stage, from the initial fixation to the final process of dissemination of the mature biofilm, is under strict regulation [62]. The QS system stands out among the main regulators of biofilm development. In strains of *S. aureus*, the QS system is responsible for the switch from biofilm formation to its disassembly [63]. Besides QS, there are several environmental influences, most notably the effects of nutrient availability, pH levels, and fluid flow, as well as regulatory systems (e.g., Sae, SarA, Rot), which are involved in the regulation of biofilm formation (Figure 4) [64].

QS allows bacteria to coordinate their behavior in a population density-dependent manner through production and accumulation of signaling molecules in the extracellular space. At a threshold concentration, these signals are recognized by bacteria and translated into changes in their profile and transcriptional behavior. Therefore, QS is responsible for bacterial synchronization, allowing for the adjustment of behavior at the population scale, akin to that in a multicellular organism. Although the basic principles of QS are conserved among a wide variety of bacteria, the signaling molecules employed for sensing differ in their structure. The acyl homoserine and lactone (AHL) QS systems represent the most-studied groups of Gram-negative bacteria, and the peptide (cyclic) QS system is the representative system in Gram-positive bacteria [65]. *S. aureus* strains utilize a QS system with autoinductive peptides (AIPs) as signaling molecules to regulate the expression of various virulence factors as well as the biosynthesis of AIPs as a function of cell population density [66].

The staphylococcal QS system, also known as the Agr system, is encoded by a 3.5 kb chromosomal locus. The locus is composed of two major transcriptional units, namely RNAII and RNAIII, which are driven by a P2 and P3 promoter, respectively. The RNAII transcript comprises the agrBDCA genes, which are responsible for encoding proteins involved in the biosynthesis, transport, signal perception and subsequent regulation of AIP target genes [67]. The signaling process occurs through a signaling cascade, starting with the transcription and translation of AgrD, the propeptide precursor of AIPs. AgrD is secreted and post-translationally modified into the final peptide by AgrB, an integral membrane endopeptidase, and further processed by SpsB type I signal peptidase [68]. At the threshold concentration in the environment, known as the quorum, AIPs trigger the binding process to the membrane-bound AgrC histidine kinase, which culminates in autophosphorylation and initiation of the signal transduction cascade [69]. Thereafter, AgrC phosphorylates the AgrA response regulator, which in turn induces the expression of RNAIII from the P3 promoter [70]. RNAIII plays a crucial role as an effector of the QS system that acts by controlling the upregulation of genes encoding secreted proteins, such as toxins and exoenzymes; it is also responsible for the downregulation of several genes that encode surface-associated adhesins [36].

A sudden increase in RNAIII levels prevents the translation of the toxin repressor (Rot) [71], which is one of the main effector molecules involved in the regulation of the QS system. Although most QS-regulated genes are regulated via RNAIII, some genes are directly regulated by AgrA. The most prominent among these is the direct regulation of RNAII from the P2 promoter, leading to elevated AIP production and a positive-feedback loop [72]. Furthermore, the two main transcripts, *psmα* and *psmβ*, in strains of *S. aureus* and *S. epidermidis,* are directly regulated by AgrA, as is the *hld* gene for the δ toxin, which is encoded in RNAIII [73].

## 8. Regulatory Genes

The formation of microbial biofilm is encoded by several biofilm-associated genes [74]. In *S. aureus*, biofilm formation is commonly encoded by 12 different genes listed in Table 1. The genes mentioned above encode different surface proteins that are involved in the adhesion of *S. aureus* cells and their penetration into the host and subsequent colonization, ultimately leading to biofilm formation and virulence. For example, in *S. aureus*, the *fib* gene is responsible for facilitating and encoding the recognition of surface fibrinogen-binding proteins, whereas the collagen-binding proteins encoded by their corresponding *cna* genes promote adhesion to the surface [75]. Autolysins are molecules responsible for adhesion, as well as cell growth and pathogenicity. In *S. aureus* isolates, AtlA, which is encoded by the *atlA* gene, is the major autolysin [76].

Intercellular adhesion can occur via an intercellular polysaccharide adhesin (IAP) through an ica-dependent (encoded by the icaADBC locus) or an ica-independent (other proteins, such as a surface protein, SasG, are involved) pathway [77,78]. The co-expression of *fnbAB* genes optimizes the penetration of *S. aureus* in the host cells, directly facilitating the formation of *S. aureus* biofilm. The *fnbA* and *fnbB* genes play the same role; however, they are not involved in the adhesion process [79].

The *clfA* and *clfB* genes are responsible for the aggregation factor, as they encode proteins anchored in the cell wall that bind fibrinogen on the host surface [80]. This binding of clustering factors A and B encoded by *clfAB* genes optimizes host colonization by *S. aureus*, promotes biofilm formation, and causes virulence through immune system evasion via the binding of soluble fibrinogen [81]. Serine-aspartate repeat factors C and D (SdrCD) facilitate attachment to desquamated epithelial cells and nasal colonization, whereas SdrE causes immune evasion by binding to complement factor H [45]. All these proteins are encoded for specific roles by the corresponding *sdr* genes. The elastin- and laminin-binding proteins encoded by their respective genes, *ebps* and *eno*, facilitate colonization of the host and biofilm formation [82].

After complete maturation, the biofilm undergoes a dispersion process, releasing the sessile cells, which can repopulate their primary site or spread to a new location, colonizing a secondary site. This dispersal of an *S. aureus* biofilm is regulated by four different genes of the Agr system [83]. The *Agr* genes encoding the dispersal of biofilm include *AgrA*, *AgrB*, *AgrC*, and AgrD [63]. Agr-regulated dispersal of biofilm occurs by the induction of different PSMs and proteases, which disperse the biofilm by acting as surfactants [84].

As its main dispersion strategy, *S. aureus* produces exoenzymes and surfactants, which play a role in the degradation of the extracellular polymeric matrix. The composition of the matrix directly reflects the effectiveness of the individual mechanisms present in the *S. aureus* strains [85]. The mechanisms that present enzymatic self-destruction pathways for proteins and/or eDNA in the matrix are less efficient in the dispersion of dependent biofilms of polysaccharides. In contrast, the mechanisms specifically targeting PIA are ineffective against polysaccharide-independent biofilms.

The secondary biofilm dispersal strategy is closely linked to the Agr quorum sensing system, involved in biofilm formation, and extracellular protease activity is required for the control of biofilm dispersal molecules [86]. Agr is expressed through bacteria present on the surface of the outer layer of the biofilm, leading to detachment and regrowth, but it is also expressed in deeper layers where it is required for channel formation [86]. This dispersion effect, linked to the Agr system, may be due to the involvement of PSMs, the expression of which is controlled by Agr quorum sensing.

Finally, nucleases, the extracellular enzymes that degrade DNA, also play an important role. Human DNaseI acts against staphylococcal biofilms and is responsible for the degradation of cellular matrices with adhered bacterial cells [87]. Staphylococcal thermonuclease nuc2 promotes dispersal in biofilm development [85]. Other nucleases, such as nuc1 nuclease, also show dispersive effects; it was reported to exhibit nuc2-like dispersion effect on biofilm in vitro [88]. The process of dispersion is complex and depends on several factors, such as bacteriophages, which have been proved to be important agents in the development of biofilm, mainly during the dispersion phase [89]. Proteases, such as Aur metalloprotease and Slp serine protease, have also been shown to be responsible for the dispersal movement.

## 9. Minimum Inhibitory Concentrations, Minimum Biofilm Inhibitory Concentrations, and Biofilm Eradication Concentrations

Before discussing the treatment of the biofilm, it is important to understand some key concepts. The biofilm-associated infections usually require a high-dose long-term antibiotic treatment, and therefore, an understanding of the antibiotic activity is important.

A specific feature of bacteria in the biofilm is their ability to survive in the presence of high doses of antibiotics [90]. Minimum inhibitory concentration (MIC) can be used as a quantitative measure of antibiotic resistance in planktonic cells [91]. The MIC and minimum bactericidal concentration (MBC) are the lowest levels of an antimicrobial agent required to inhibit growth and to kill a particular bacterium, respectively [92]. Minimal biofilm inhibition concentration (MBIC) and minimal biofilm eradication concentration (MBEC) are based on the same premise, but refer to the concentrations relevant for cells in a biofilm [93]. The MIC is much higher for bacteria that form biofilm compared to that for bacteria that do not [94]. This concurs with the observation that biofilms are resistant to antibiotic concentrations up to 1000× greater than those required to kill free-living bacteria [18] and signifies a pressing need for combination therapy instead of monotherapy. The emergence of *S. aureus* isolates that are resistant to multiple antibiotics is a real concern, especially as it is exaggerated among MRSA strains [95].

Sensitivity tests are necessary for an appropriate choice and dose of antimicrobial therapy. MIC and MBEC of bacteria are variables that help in reducing the spread of resistant strains and direct the treatment. Staphylococcal isolates from biofilm show a much higher breakpoint for MBEC than for MIC, indicating the importance of applying both the biofilm susceptibility tests [96]. Although the MBEC and MIC of vancomycin (VAN) in planktonic cells are similar, they are markedly different for biofilm-producing isolates; therefore, from a clinical perspective, MBEC is the preferred measure [97]. Despite the availability of standardized methods to treat biofilm, most successful approaches were tested on planktonic cells. Although MBEC and MBIC values are proposed, this is confounded by limited evidence and complexity of correlation between innate activity toward planktonic cells and those in a biofilm [98]. A summary of the main conventional antibiotic treatment of *S. aureus* biofilms is presented in Table 2.

## 10. Biofilm Treatment

The ineffectiveness of antibiotics against infections caused by bacterial biofilms poses a major challenge in infection microbiology [99]. *S. aureus* infections associated with biofilms are difficult to eradicate because of the high tolerance of this bacterium to antibacterial agents and to host immune defenses [17]. The treatment of such infections using conventional antibiotic therapy is challenging as only doses that are sublethal to the biofilm can be administered safely to patients [100]. The eradication of these biofilm infections is complicated because in biofilms bacteria cells are encased in a self-produced extracellular matrix composed of proteins, polysaccharides, and extracellular DNA, which protects them against the host immune system and antimicrobial agents. Moreover, bacteria in biofilms may enter a low metabolic state, which dramatically increases their tolerance to antibiotics [101]. Consequently, bacterial cells in biofilms may tolerate up to 1000 times higher concentrations of antibiotics than their planktonic counterparts [102]. Thus, antibiotics cannot be dosed at a concentration sufficient to eradicate the biofilm without causing detrimental side effects to the patient. The only recourse is surgical removal of the biofilm, which is costly and, in some cases, not feasible [103].

A significant problem often associated with *S. aureus* infections is the rapid development of antibiotic resistance. In *S. aureus* biofilm infections, this may be compounded by an increase in MICs compared with that for planktonic isogenic bacteria, indicating antibiotic tolerance [94]. In addition, exposure of increased numbers of *S. aureus* cells in a biofilm to the antibiotic selection pressure is also associated with the potential development of antibiotic resistance. Antibiotic treatment of biofilm-associated infection may also result in the development of dormant “persister” populations of cells that can withstand the treatment. Therefore, biofilm-related infection (a chronic infection) is now defined as one that persists despite antibiotic treatment and innate and adaptive immune responses of the host and is characterized by a persistent pathology. Once the administration of antibiotics is stopped, most patients (>80%) have a recurrence of infection [103].

The clinical implications of microorganisms growing as biofilms are that they may be more difficult to recover from clinical samples, and that they are physiologically much more resistant to the effects of antibiotics and disinfectants [102]. Moreover, antibiotic therapy based on susceptibility testing of planktonic (nonaggregated) microorganisms may be associated with treatment failure or recurrence of the infection [103]. Antibiofilm strategies are mainly of two types—those involving the inhibition or prevention of new biofilm formation and those based on dispersal or eradication of existing biofilms [17].

*S. aureus* cells within a complex biofilm matrix are refractory to both systemic antimicrobial agents and host immune responses [104,105]. Treatment of biofilm infection requires sensitive and well-penetrating antibiotics to ensure a sufficient concentration of effective antibiotics at the site of biofilm infection. Hence, tetracyclines, macrolides, rifamycins, lincosamides, quinolones, fusidic acid, oxazolidinones, sulfonamides, and nitroimidazole are preferred to glycopeptides, aminoglycosides, polymyxins, and β-lactamases because they can penetrate deeper [106]. In addition to the biofilm age and level of resistance to a given antibiotic, broader considerations for treatment include appropriate duration of antibiotic regimen and dosage optimization [107]. Bacteria within a biofilm can exhibit resistance to multiple treatments, even in the presence of high concentrations of bactericidal and bacteriostatic antibiotics and toxic compounds, in stark contrast to those in their planktonic form. Among the various mechanisms by which this complex phenomenon may occur, those involving antibiotic efflux, enzyme activity, and reduced permeability are noteworthy. The mechanisms by which antibiotics inhibit and or disrupt *S. aureus* biofilm are not fully known and have not been reported yet. Therefore, the antibiofilm mechanisms of antibiotics remain an area that needs to be explored for devising effective therapeutic strategies against *S. aureus* biofilm-related infections.

### 10.1. Glycopeptides

VAN is the most commonly administered drug for treating *S. aureus* biofilm-associated infections [17]. However, prolonged, persistent, or recurrent bacteremia during therapy; high rates of microbiological and clinical failures; nephrotoxicity; and increasing prevalence of nonsusceptible strains limit its use in several infections [108,109]. Previous studies have also found that VAN does not fully penetrate staphylococcal biofilms [110,111]. Considering that VAN is still the drug of choice for MRSA infections, it is the most studied drug in biofilm-associated infections in this pathogen, as well as for MSSA [112]. VAN has been extensively studied in MSSA because the low activity of beta-lactams in biofilms is known, regardless of their diffusion [113].

VAN can inhibit biofilm production, presenting a low MBIC, but an extremely high MBEC. Douthit et al. suggested that VAN can effectively eradicate biofilm, but only at concentrations greater than 6000 mg/L, which are impossible to achieve via the systemic route [114]. The high MBEC for VAN has been confirmed in other studies and has always been found to be 1000 to 4000 times above the MIC [115,116]. The variability in the response of VAN to antibiofilm activity is dependent on biofilm maturation, with the activity being higher in younger than in mature biofilms. Moreover, in in vitro studies, the activity is better than in in vivo studies, suggesting that variants related to pharmacokinetics are important for the success of therapy [115,116]. In in vitro studies, the exposure time is another fact that interferes with the response of *S. aureus* biofilm to VAN [117].

Data on VAN present in the literature should not be extrapolated to other glycopeptides because telavancin, for example, has better antibiofilm activity [118]. A study with dalbavancin also showed better antibiofilm activity than VAN [119].

### 10.2. Penicillins

MSSA bacteremia is generally treated with a beta-lactam agent, such as nafcillin, oxacillin, flucloxacillin, or cefazolin. Currently, antistaphylococcal penicillins, such as nafcillin and oxacillin, are recommended as first-line agents in the treatment of MSSA infections, with cefazolin reserved as an alternative for patients intolerant to these agents or for dosing convenience (e.g., for outpatient parenteral antibiotic therapy or hemodialysis) [120].

Oxacillin, like other beta-lactam antibiotics, binds penicillin-binding proteins (PBPs) that weaken or interfere with cell wall formation. After binding to PBPs, the cell wall weakens or undergoes lysis. This drug acts in a time-dependent manner (i.e., it is more effective when drug concentrations are maintained above the MIC during the dose interval). Oxacillin has a limited spectrum of activity that primarily includes Gram-positive bacteria. Staphylococci are susceptible to oxacillin because it is resistant to the beta-lactamase produced by *Staphylococcus* spp. Data on the effect of oxacillin administered during the early stage of on biofilm formation or on preformed mature biofilm are scarce. Mirani et al. demonstrated the effect of this antibiotic on *S. aureus* reference strains [121]. The mechanism underlying the observed effect could be the modulation of the *icaA* and *agr* expression, the two major regulator genes in biofilm formation. Manner et al. confirmed the efficacy of oxacillin on mature biofilm [122].

In MSSA biofilms, the MBEC increases 4 to 100 times. This concentration is impossible to achieve pharmacologically, considering the need for systemic infusion of the drug, suggesting poor drug efficacy against biofilm-associated infections [123]. In MBEC tests with oxacillin, using isolates with MIC = 0.25 mg/L, MBEC reached 128 mg/L, which is an extremely high concentration, proving that oxacillin is an inadequate antibiotic for the treatment of infections associated with biofilms [124].

### 10.3. Rifampin

Rifampin (RIF) is the main drug for treatment of infections of mycobacterial origin, such as those caused by *Mycobacterium bovis* and *Mycobacterium leprae*; however, it can also be used in Gram-positive infections, as well as in MRSA strains [125]. The antibiotic acts by inhibiting the synthesis of DNA-dependent RNA polymerase [126]. Its effect has been investigated in models of orthopedic device-related infections. In osteoblast infection models, two specific RIFs, rifapentine and rifabutin, consistently reduced biofilm-embedded bacteria for all *S. aureus* isolates [127]. On a cautionary note, RIF should be considered for enhanced antistaphylococcal activity but should not be used alone [128]. Despite the overwhelming evidence for the antibiofilm activity of RIF, there are a few studies in which no beneficial effect of RIF was observed [129,130,131]. RIF is the only traditionally administered antibiotic with high and reliable antibiofilm activity against *S. aureus* [132]. However, RIF monotreatment is avoided clinically because *S. aureus* can develop rapid resistance to this drug. In periprosthetic joint infections, RIF is only used in combination with another antibiotic, such as cefazolin or VAN [133,134]. Nevertheless, RIF remains the only antibiotic that is highly efficacy against biofilm-associated staphylococci, and when used in combination with other antibiotics, it represents the best currently available treatment along with surgical debridement with retention for treating prosthetic joint infections [132].

The guidelines’ recommendation to combine RIF with other antibiotics is based on several in vitro studies. RIF is capable of reducing the MBEC of different antibiotics, which would not have any antimicrobial activity and do not cross the polysaccharide barrier of *S. aureus* biofilms. RIF can reduce the MBEC of gentamicin, VAN, and cefazolin [115]. However, studies on monotherapy with RIF have shown failure in eradicating biofilms [124].

### 10.4. Aminoglycosides

Amikacin (AMK) and gentamicin are the most important antibiotics representing the group of aminoglycosides and are characterized by their effect on most Gram-negative bacteria [135]. The mechanism of action of these antibiotics involves binding to specific proteins of the 30S and 16S rRNA subunit, thereby inhibiting protein synthesis and assembly of the initiation complex related to mRNA [136]. They are effective against both MSSA and MRSA. Although clinical data are lacking, aminoglycosides, used either in a monotherapy or in combination with rifampicin, have better antibiofilm activity than other antibiotics, for example, beta-lactams, at least in vitro. When used in combination with rifampicin, the MBEC of aminoglycosides can be brought to therapeutic concentrations, and they can potentially be used for treatment of catheter biofilms, endocarditis, and infections related to orthopedic implants [115]. In another study, the same benefits of aminoglycosides were not observed for MRSA and MSSA isolates, indicating the dependence on the tested strain [137].

In an in vivo model of implant biofilm, gentamicin showed improved activity when compared with that of VAN; the activity reached 100% in 48 h [138]. In the same study, considering the low efficacy of ceftriaxone against biofilms, this drug was combined with tobramycin (another aminoglycoside). This combination exhibited greater antibiofilm activity than that of the antibiotics when used alone. In another static biofilm model, monotherapy with gentamicin was proven to be the best compared with monotherapies with clindamycin, linezolid, and VAN. The combination of bacteriostatic antibiotics with gentamicin appeared to have antagonistic effects. When compared with daptomycin (DAP), gentamicin showed antibiofilm activity but required higher concentrations [139].

### 10.5. Cephalosporins

Cephalosporins are antimicrobials belonging to the class of beta-lactams. They are widely applicable and are active against both Gram-positive and Gram-negative bacteria. In clinical practice, cephalosporins have wide applicability, mainly against resistant bacteria, central nervous system infections, and skin infections [140]. The antimicrobial action of cephalosporins involves binding to and inactivation of PBPs; however, owing to the diversity of classes, different cephalosporins have a different affinity for these proteins [141]. These antimicrobials are frequently used alone or in combination with an aminoglycoside for empiric therapy of nosocomial infections. The effect of protein binding on antimicrobial efficacy has been a subject of intense debate. Nonetheless, there is considerable evidence from in vitro studies and in vivo animal models indicating that the presence of serum proteins significantly impairs the activity of highly protein-bound beta-lactams against *S. aureus* [142,143,144]. Two cephalosporins, ceftobiprole and ceftaroline, have been shown to be clinically effective in the treatment of MRSA skin and soft infections [145].

In vitro data indicate that cefazolin may sometimes be subject to an inoculum effect, where it can be hydrolyzed by increased production of beta-lactamases [146]. This effect is defined as a significant increase in the cefazolin MIC at high inoculum (10^7^ colony-forming units (CFU)/mL) compared with that at standard inoculum (10^5^ CFU/mL) [147,148]. The clinical significance of this phenomenon for biofilms is uncertain; studies examining the significance of this effect have produced conflicting results [149]. In addition, a study on the use of cefazolin in biofilms demonstrated a very large increase in MIC, generating MBEC 1000 to 2000 times greater, making it an ineffective drug for use against biofilms [115].

Although *S. aureus* isolates from patients with endocarditis were reported to be susceptible to ceftriaxone in vitro, this antibiotic had no antibiofilm activity [150]. This could also be demonstrated when the drug was used as a systemic monotherapy in an in vivo model of implant-related infection. However, when used in combination with aminoglycosides, the effect was improved, although 100% success was not achieved [138].

### 10.6. Clindamycin

Owing to its effects against anaerobic germs, clindamycin can be used in intra-abdominal infections and uncomplicated skin infections under several scenarios; the antibiotic also exhibits potential prophylactic effects [151]. Clindamycin has also been reported to be effective in MRSA infections [152]. It acts as a protein synthesis inhibitor that binds to the 50S subunit of the bacterial ribosome [153]. Previous in vitro studies highlight the need to adjust the antibiotic dose, given that subinhibitory concentrations modulate the composition of the biofilm matrix, impacting the autolysis of matrix component cells and the release of extracellular DNA, PSMs, and FnbBs, which are closely linked to cell adhesion factors and ensure a biofilm with greater stability [154]. The applicability of clindamycin extends to orthopedic infections, especially biofilm osteomyelitis, because it has good bone penetration and shows a rapid bacteriological response. Besides being a cheap option, it allows for a quick oral switch with few side effects and is also safe for the pediatric population [152].

Clindamycin has been widely used in the treatment of orthopedic infections owing to its penetration into bone and soft tissues. On the contrary, in an in vitro study, it was shown to have scant antibiofilm activity, even when combined with the activities of other antibiotics, such as linezolid, gentamicin, and VAN [139].

### 10.7. Daptomycin

DAP, a cyclic lipopeptide molecule, is a novel antibiotic that has been used for VAN-unresponsive *S. aureus* infections. It disrupts the cytoplasmic membrane of bacteria, resulting in rapid depolarization and cessation of DNA, RNA, and protein synthesis. Among four drugs tested (linezolid, clindamycin, VAN, and tigecycline), DAP was found to be the most effective in clearing *S. aureus* from an existing biofilm, and it provides an alternative treatment option for MRSA and VRSA, effectively targeting biofilms [155]. This antibiotic has been shown to be highly effective against biofilms of a panel of MRSA clinical isolates [155,156]. However, a small population of biofilm bacteria remained tolerant to DAP, and ambiguous results were obtained when this drug was used in combination with other antibiotics [157,158].

Randomized clinical trials of alternative agents, such as DAP, show that it is comparable, or more precisely, non-inferior, but not superior, to standard therapy [108,159,160,161]. Quantitative in vitro studies using biofilms of MRSA strains demonstrated the effect of DAP as a bactericidal agent against resistant bacteria, both in suspended form and in adherent biofilm organizations. Even in a resistant strain scenario, the MBEC values of DAP were lower than those of the other antibiotics tested, and the antibiotic was better than gentamicin and tigecycline [162]. There is evidence confirming a model of action, in which lower concentrations to reach the MBEC were obtained, being up to 4 times lower compared with those of tigecycline and other antibiotics, with DAP and RIF performing better in different sensitivity and eradication tests [163]. In studies aimed at evaluating bactericidal effects in comparison models of cell survival associated with biofilms in MRSA strains, DAP showed a rapid bactericidal effect and better performance compared with other drugs, such as tigecycline, linezolid, VAN, and clindamycin. In cell survival comparison studies, DAP could eliminate an average of 96% of cells from the formed films, and the proportion of surviving cells was a smaller in case of exposure to DAP compared with that in the case of all the other antibiotics used [155,162,163].

### 10.8. Doxycycline

Recently, suppressive doxycycline therapy for *S. aureus* was reported in a small and high-risk prosthetic joint infection group. The cohort showed reasonable effectiveness and tolerability of the antibiotic for successful treatment [164]. Reports on the use of doxycycline for *S. aureus* biofilms are scarce, although it has been part of endeavors studying multiple schemes. Although doxycycline has a role in modifying the QS of Gram-negative bacilli, this activity is speculated to be exerted on Gram-positive bacilli as well [165]. Doxycycline has an inhibitory effect on biofilms, both in respect to biomass and the production of surface polysaccharides, which are the main components of biofilms [166]. Monotherapy may not be effective and even combinations with other antibiotics have discreet effect [167].

### 10.9. Linezolid

Linezolid acts by inhibition of bacterial protein synthesis by binding to 23S rRNA in the catalytic site of the 50S ribosome [168]. Parra-Ruiz et al. (2012) reported that linezolid, when used alone, was ineffective in reducing the *S. aureus* biofilm [169]. Gander et al. observed an effect of 1× MIC on the early stage of biofilm formation, but they used a classic microbiological medium and static conditions [170].

The *ica* gene of *S. aureus* is vital for its growth and biofilm formation. IcaA and IcaB are critical proteins in the synthesis of extracellular polysaccharides and in the formation of *S. aureus* biofilms [171]. Linezolid causes invagination of the *S. aureus* cell surface and inhibits the production of biofilms.

Linezolid exhibits interesting penetration and antibiofilm activity in in vitro studies as well as in molecular studies performed to elucidate its mechanism of action. A study comparing the efficacy of antibiofilm drugs against MRSA revealed that linezolid has reasonable activity against biofilms, although the activity was lower than that of DAP. The penetration of linezolid was as high as that of daptomycin, but the latter had better inhibitory activity [110].

### 10.10. Ertapenem

Ertapenem is a once-a-day parenteral β-lactam antimicrobial that can be used in monotherapy for the treatment of various community-acquired infections, including community-acquired pneumonia; acute pelvic infection; and complicated intra-abdominal, skin, and urinary tract infections. There is a correlation between exposure duration and penetration time. Jefferson et al. found that the penetration time of antibiotics in biofilms ranges from a few minutes to almost 24 h [111]. Oxacillin, cefotaxime, VAN, and delafloxacin are antibiotics with limited penetration in staphylococcal biofilms, whereas other antibiotics, such as amikacin and ciprofloxacin, are unaffected by the presence of these biofilms [172]. For all antibiotics, concentrations around bacterial cells in deeper layers are gradually increasing and an early exposure to subinhibitory concentrations can favor the entrance in the persister state, thereby making these bacteria survive subsequent lethal concentrations [173]. *S. aureus* biofilms have been shown to become more susceptible to antibiotics with increased exposure time from 1 to 5 days [117]. The molecular weight of antimicrobial agents has been suggested to have a low impact on biofilm permeability. VAN, teicoplanin, DAP, and arbekacin are positively charged under physiological conditions, and DAP and arbekacin show high biofilm permeability. Therefore, factors other than the charge of antimicrobial agents, such as polarity, may also affect biofilm permeability [174].

## 11. Combination Antibiotic Therapy for *S. aureus* Biofilms

VAN is the most commonly administered drug for *S. aureus* biofilm-associated infections; however, increased tolerance of biofilms to VAN (planktonic MIC ~2 μg/mL, biofilm MIC ~20 μg/mL) warrants the use of a combination of other drugs. VAN and RIF combinations have been studied, particularly in the context of biofilm infections. The efficacy of the combination of VAN and RIF is due to a reduction in bacterial adhesion [175]. However, conflicting results have been reported for this combination. Multiple studies indicate that, although this combination might be effective against MSSA, it may not hold promise for the treatment of MRSA biofilm infections [132,176]. Rose and Poppens (2009) demonstrated that the reduced bacterial killing in high-biofilm-producing *S. aureus* by VAN was overcome with the addition of tigecycline or RIF. Utilizing these agents at 4× MIC, in combination with 15 mg/L VAN, bactericidal activity was achieved by 24 h against all isolates with similar activity. A combination of RIF and VAN caused an average reduction of 4.6 CFU/mL from the initial inoculum, whereas a reduction of 4.3 CFU/mL was observed with tigecycline plus VAN [177].

A recent in vitro study proved the effectiveness of the combination of VAN and AMK against planktonic cells, however, without any effect on bacteria incorporated in the biofilm. The synergism occurs through the action of VAN as a reducing factor in the selection of bacteria with lower susceptibility to AMK [178].

Recently, several studies have confirmed synergistic bactericidal effects of combinations of VAN and FOS on *S. aureus* in vitro [179,180] or in multiple MRSA strains embedded in biofilms [181,182]. However, these results have not been confirmed in vivo [183].

Combinations of RIF with other antibiotics represent the best treatment currently available besides surgical debridement with retention for treating prosthetic joint infections [128,132,184]. RIF also reportedly shows antibiofilm activity in combination with another antibiotic, ciprofloxacin, against *S. aureus* biofilm [185]. Yang et al. demonstrated that a combination of RIF with ceftiofur and that of RIF with doxycycline had an interactive effect against *S. aureus*. However, kanamycin and lincomycin showed antagonistic activities when used in combination with RIF [186].

Ambiguous results have been reported for combinations of DAP with other antibiotics [158]. To assess its therapeutic synergism with FOS, a randomized clinical trial was performed, wherein outcomes were evaluated after 6 weeks of successful clinical treatment against MRSA strains in patients with endocarditis. A combination of the antibiotics resulted in a 12% higher success rate when compared to a single DAP therapy, although the results were not statistically significant. In addition, the preventive association was higher against microbiological failure and bacteremia but was associated with a higher frequency of side effects leading to the discontinuation of therapy [187]. FOS also has synergistic effects with other antibiotics in the treatment of MRSA, *Streptococcus*, *Enterococcus*, and Enterobacteriaceae species [187,188,189].

Parra-Ruiz et al. (2012) demonstrated that a combination therapy of linezolid plus DAP significantly improved the bacterial killing effect of both the agents against the biofilms of MRSA with sustained bactericidal activity and absolute bacterial count reductions >5 log10 CFU/mL [169]. Hu et al. (2019) reported the use of azithromycin and clindamycin for the inhibition and dispersal *S. aureus* biofilm. The antibiofilm mechanism of azithromycin is based on its ability to disrupt bacterial QS [190].

## 12. Future Perspectives

In addition to conventional antibiofilm therapeutic strategies, a wide range of substances with different mechanisms to prevent or eradicate staphylococcal biofilms has been identified, but none of effective current clinical application. Among these substances are molecules that interfere with the Agr QS system and degrade the biofilm matrix, or have bactericidal activity against bacterial cells within biofilms (e.g., lysostaphin) [191]. Another highly promising strategy is to modify surfaces of biomaterials, such as catheters and implants, to prevent staphylococcal adhesion and subsequent biofilm formation. In this regard, implants coated with surfactants, antibiotics, or antimicrobial peptides alone or in combination have shown promising effects [192]. Another interesting new strategy to combat staphylococcal biofilms is the application of nanoparticles as delivery systems for conventional antibiotics or in the form of metallic nanoparticles with antimicrobial activity, mainly based on silver nanoparticles [193,194]. Other nanoparticles have been tested for delivery of antibiotics to inhibit *S. aureus* biofilm formation. Liposomes have similarly shown promising delivery vehicles for antibiotics [195].

The true fact is that in therapeutics against biofilms, we still need robust clinical studies to show the effectiveness of these innovations.

## 13. Conclusions

Treatment guidelines are specific for the type of infection, and for *S. aureus*, they depend upon antibiotic susceptibility. These clinical practice guidelines are based on randomized controlled clinical trials, comparing existing and novel treatments, to determine optimal antibiotic regimens and are often debated within the clinical community; however, not all include guidelines for biofilm infections. A majority of chronic staphylococcal infections are now recognized to be due to biofilms, particularly those associated with an indwelling medical device. However, most therapeutic strategies are applicable only to planktonic or acute *S. aureus* infections. In the first published guideline for biofilm infections, the use of frequent, appropriate, empiric antibiotic therapy is recommended for *S. aureus*, especially in institutions with recurring MRSA infections. Therefore, there is an urgent unmet need for new therapeutic strategies that target *S. aureus* in biofilms.

Future perspectives in the treatment of infections associated with biofilms are clinical and controlled studies with currently available antibiotics. There is much research into alternative treatments, such as enzymes to break the saccharide chains, proteinases, and DNases. However, these studies are still in their infancy, and their main challenge is to make them available at the site of infection without toxicity.

## Figures and Tables

**Figure 1 antibiotics-12-00087-f001:**
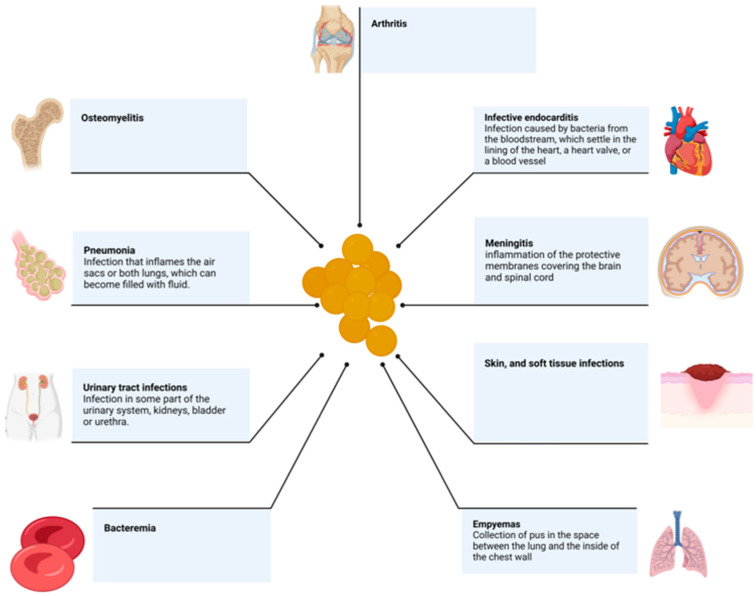
Schematic examples of the types of *Staphylococcus aureus* infections.

**Figure 2 antibiotics-12-00087-f002:**
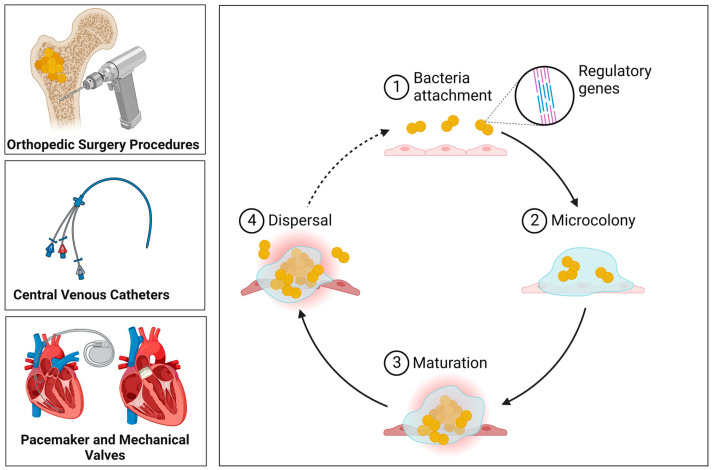
Development of staphylococcal biofilm.

**Figure 3 antibiotics-12-00087-f003:**
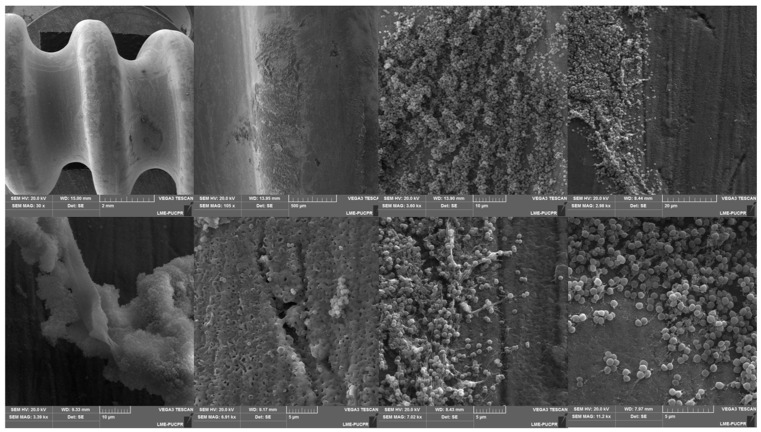
Development of *Staphylococcus aureus* biofilm on medical devices (orthopedic titanium screw).

**Figure 4 antibiotics-12-00087-f004:**
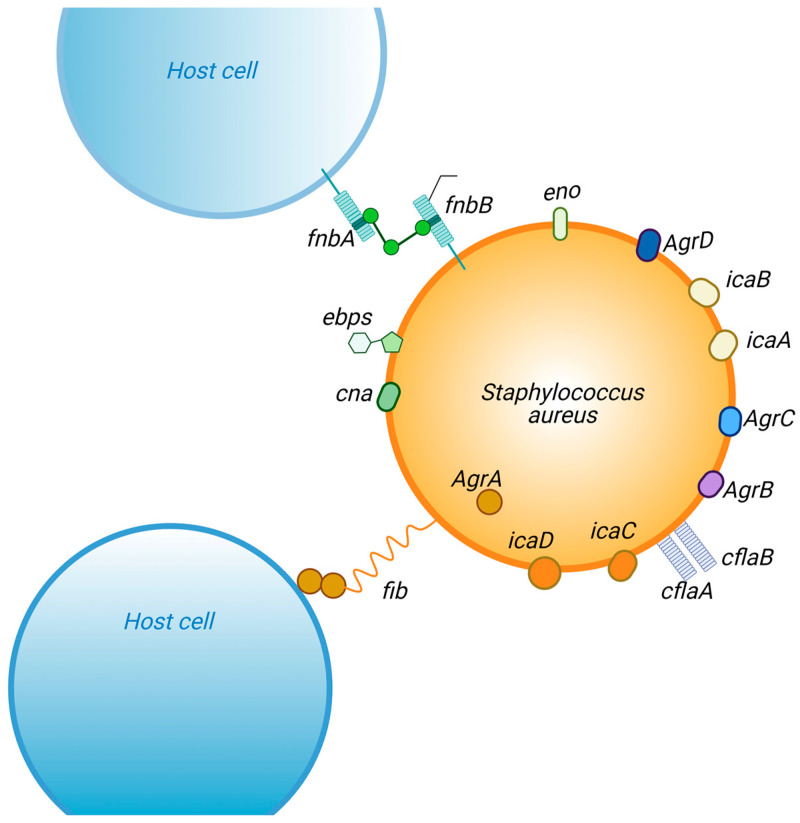
Genes associated involved in the regulation of *S. aureus* biofilm formation.

**Table 1 antibiotics-12-00087-t001:** Different genes and their corresponding encoding mechanisms in *Staphylococcus aureus* biofilm formation and maturation.

	Gene(s)	Mechanism
biofilm formation	Fibrinogen-binding proteins (fib)	recognition of surface fibrinogen-binding proteins
	Fibronectin-binding proteins (fnbA and fnbB)	Invasion of host cells
	intercellular adhesion (icaA, B, C and D)	Adherence/intracellular atachment
	clumping fator A and B (clfA and clfB)	Adherence/intracellular atachment
	Elastin-binding protein (ebps)	colonization
	Laminin-binding protein (eno)	colonization
	Collagen-binding protein (cna)	Adherence/intracellular atachment
biofilm maturation	accessory gene regulatory A (AgrA)	triggering the intracellular communication
	accessory gene regulatory B (AgrB)	regulate the expression and transportation of autoinducing peptide
	accessory gene regulatory C (AgrC)	regulate the expression and transportation of autoinducing peptide
	accessory gene regulatory D (AgrD)	triggering the intracellular communication

**Table 2 antibiotics-12-00087-t002:** Conventional antibiotic treatment of *Staphylococcus aureus* biofilms.

Class	Compound	Principle	Aplication
Cephalosporins	Cefazolin	Cell wall synthesis inhibition	Osteomyelitis
			skin and soft tissue infections
	Cephalexin		Simple, uncomplicated skin infections
			Osteomyelitis
	Cefaclor		skin infections
	Cefotaxime		skin infections
	Ceftriaxone		bloodstream infections
Fluoroquinolones	Moxifloxacin	DNA synthesis inhibitors	endocarditis
	Ciprofloxacin		endocarditis
	Delafloxacin		skin and soft tissue infections
Glycopeptides	Vancomycin	Cell wall synthesis inhibition	Complex skin and soft-tissue infections
			Bacteremia
			Catheter-related infections
			Osteomyelitis
			Pneumonia
	Teicoplanin		endocarditis
Lincosamide	Clindamycin	Protein synthesis inhibitors	Simple, uncomplicated skin infections
			Osteomyelitis
Lipopeptides	Daptomycin	disrupts the cytoplasmic membrane of bacteria, resulting in rapid depolarization and cessation of DNA, RNA, and protein synthesis	Bacteremia and Endocarditis
Macrolides	Erythromycin	Protein synthesis inhibitors	skin infections
Miscellaneous agents	Fosfomycin	Cell wall synthesis inhibition	diabetic patients presenting with bacterial foot infection
	Trimethoprim/sulfamethoxazole	inhibiting an essential step in the synthesis of bacterial nucleic acids and proteins	endocarditis
			bone and joint infections
			meningitis
Penicillins	Nafcillin, Dicloxacillin, Amoxicillin-clavulanate, Ampicillin-sulbactam	Cell wall synthesis inhibition	Complex skin and soft-tissue infections
			Bacteremia
			Catheter-related infections
			Osteomyelitis
			Pneumonia
Rifamycin	Rifampin	RNA synthesis inhibitors	orthopedic implant
Tetracyclines	Doxycycline	Protein synthesis inhibition. Anti-30S ribosomal subunit	
Glycylcycline	Tigecycline	Protein synthesis inhibition Anti-30S ribosomal subunit	
Aminoglycosides	Amikacin	Protein synthesis inhibitors	in combination with fosfomycin

## Data Availability

Not applicable.

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
