# Peer review of "Antimicrobial Treatment of Staphylococcus aureus Biofilms"

_antibiotics, 2023, doi:10.3390/antibiotics12010087_

Round 1

Reviewer 1 Report

In this article, the authors detail the genomic entities involved in biofilm formation, quorum sensing, and antimicrobial activity in monotherapy and combination therapy. They have also discussed various treatment options for S. aureus infections, and the need for new therapeutics to treat S. aureus in biofilms. The manuscript is nicely organized and well-written, however, the authors need to address some minor comments to improve the manuscript. The authors need to check the usage of italics and spacing throughout the manuscript.

Minor Comments:

L115 à move it to L114 and remove paragraphing

Table 1: Please re-write the title

L454, L467: in vitro à italics. Please check throughout the manuscript

L507: in vivo à italics. Please check this throughout the manuscript

Author Response

Author's Reply to the Review Report (Reviewer 1)

In this article, the authors detail the genomic entities involved in biofilm formation, quorum sensing, and antimicrobial activity in monotherapy and combination therapy. They have also discussed various treatment options for S. aureus infections, and the need for new therapeutics to treat S. aureus in biofilms. The manuscript is nicely organized and well-written, however, the authors need to address some minor comments to improve the manuscript. The authors need to check the usage of italics and spacing throughout the manuscript.

Minor Comments:

L115 à move it to L114 and remove paragraphing

ANSWER: Ok

Table 1: Please re-write the title

ANSWER: Ok

L454, L467: in vitro à italics. Please check throughout the manuscript

ANSWER: Ok

L507: in vivo à italics. Please check this throughout the manuscript

ANSWER: Ok

Reviewer 2 Report

Please find my comments in the attached fle

Author Response

Author's Reply to the Review Report (Reviewer 2)

Reviewer comments: Title: Antimicrobial treatment of Staphylococcus aureus biofilms Although some related reviews have been published, the present review is wellwritten, and could be published if the authors can fulfil the following points.

  1. Keywords: Please check keywords initials

ANSWER: Ok

  1. Introduction:
  2. In the introduction section: you need to show the efforts of previous researchers and what is new in your review. At the end of the introduction section, please summarize your objectives and what make your review is worse publishing although related reviews [1-3] were recently published: 1 Bhattacharya, M., Wozniak, D.J., Stoodley, P. and Hall-Stoodley, L., 2015. Prevention and treatment of Staphylococcus aureus biofilms. Expert review of anti-infective therapy, 13(12), pp.1499-1516. 2 Suresh, M.K., Biswas, R. and Biswas, L., 2019. An update on recent developments in the prevention and treatment of Staphylococcus aureus biofilms. International Journal of Medical Microbiology, 309(1), pp.1-12. 3 Idrees, M., Sawant, S., Karodia, N. and Rahman, A., 2021. Staphylococcus aureus biofilm: Morphology, genetics, pathogenesis and treatment strategies. International Journal of Environmental Research and Public Health, 18(14), p.7602.

ANSWER: Thank you very much. We have included a justification for our review.  

  1. Figures, the resolution oof most of figures especially figure 1 is very low.

ANSWER: Thank you. We have improved the size of the figure and resolution.  

  1. Please be careful of writing microbial genus and species name in italic.

ANSWER: Thank you. We have checked all names in the manuscript.  

  1. The review lack the strategy and source of data collection: In this section you need to provide the source of data (websites), the keywords for the search, your strategy, the covered period,…..etc.

ANSWER: This is not a systematic review, it is a narrative review, this is the subject to avoid the search strategy in the manuscript.  

  1. Authors need to provide some future perspectives?

ANSWER: We have included in the last paragraph of the manuscript a brief future perspective

Reviewer 3 Report

The manuscript of “Antimicrobial treatment of Staphylococcus aureus biofilmsis interesting to read and appreciated for good attempt. Author explained about several things like, biofilm formation, quorum sensing, and anti-14 microbial activity in monotherapy and combination therapy. I decided minor revision and also correct the following suggestion before accept the manuscript.

1.      The abstract did not fulfill all the results and it should be written corrected with obtained results.

2.      Keywords should be in order and relate to the manuscript

3.      Hypothesis of introduction part is very limited. Need current explanation.

4.      In table lot of general issues are discussed, images quality is low and the very normal.

5.      Some many references are old like 15 years ago, try t to add new references.

Author Response

Author's Reply to the Review Report (Reviewer 3)

The manuscript of “Antimicrobial treatment of Staphylococcus aureus biofilms” is interesting to read and appreciated for good attempt. Author explained about several things like, biofilm formation, quorum sensing, and anti-14 microbial activity in monotherapy and combination therapy. I decided minor revision and also correct the following suggestion before accept the manuscript.

  1. The abstract did not fulfill all the results and it should be written corrected with obtained results.

ANSWER: The manuscript is a narrative review. All the results of the review, I believe that is not possible to include in the abstract. However, we would be grateful if the reviewer suggests any items that are relevant for inclusion.

  1. Keywords should be in order and relate to the manuscript

ANSWER: The Keywords are related to the manuscript and in the alphabetic order. If the referee suggest in order of relevance, we can change it.

  1. Hypothesis of introduction part is very limited. Need current explanation.

ANSWER: The manuscript is a narrative review. A specific hypothesis to be tested is no possible beyond the extensive discussion about several genes and antibiotics used in the treatment of infections associated with biofilms. We have included a better justificative to support the review.

  1. In table lot of general issues are discussed, images quality is low and the very normal.

 ANSWER: We have improved the quality of images. The tables summarize info from the text for the readers.

  1. Some many references are old like 15 years ago, try t to add new references.

 ANSWER: From 190 references, 39 references are old 15 years ago, and we revised each one. These references represents original articles about therapy and genetic issues, which could not be substituted.

Reviewer 4 Report

Dear Authors,

I reviewd your proposed manuscript entitled "Antimicrobial treatment of Staphylococcus aureus biofilms". The authors focus on a broad discussion on the genes involved in biofilm formation, quorum sensing, and antimicrobial activity in monotherapy and combination therapy as facilitatind tools that should benefit researchers engaged in optimizing the treatment of infections associated with S. aureus biofilms.

The manuscript is well written and packs an important number of recent findings. I suggest including a section about developments made in medical devices biotechnology in order to combat biofilm formation. From modifying the surphaces interaction with biofilms to composites used for interfering with adhesion, etc.
As well as a section with alternative treatments beside antibiotics.

Good luck!

Author Response

Author's Reply to the Review Report (Reviewer 4)

Comments and Suggestions for Authors

Dear Authors,

I reviewed your proposed manuscript entitled "Antimicrobial treatment of Staphylococcus aureus biofilms". The authors focus on a broad discussion on the genes involved in biofilm formation, quorum sensing, and antimicrobial activity in monotherapy and combination therapy as facilitating tools that should benefit researchers engaged in optimizing the treatment of infections associated with S. aureus biofilms.

The manuscript is well written and packs an important number of recent findings. I suggest including a section about developments made in medical devices biotechnology in order to combat biofilm formation. From modifying the surphaces interaction with biofilms to composites used for interfering with adhesion, etc.
As well as a section with alternative treatments beside antibiotics.

Good luck!

 ANSWER: Thank you very much. We have included a section about this issue before conclusion.